# Pathways Related to NLRP3 Inflammasome Activation Induced by Gold Nanorods

**DOI:** 10.3390/ijms23105763

**Published:** 2022-05-20

**Authors:** Rob J. Vandebriel, Sylvie Remy, Jolanda P. Vermeulen, Evelien G. E. Hurkmans, Kirsten Kevenaar, Neus G. Bastús, Beatriz Pelaz, Mahmoud G. Soliman, Victor F. Puntes, Wolfgang J. Parak, Jeroen L. A. Pennings, Inge Nelissen

**Affiliations:** 1Centre for Health Protection, National Institute for Public Health & the Environment, 3720 BA Bilthoven, The Netherlands; jolanda.vermeulen@rivm.nl (J.P.V.); evelienhurkmans@hotmail.com (E.G.E.H.); kirstenkevenaar@hotmail.com (K.K.); jeroen.pennings@rivm.nl (J.L.A.P.); 2Health Unit, VITO NV, 2400 Mol, Belgium; sylvie.remy@vito.be (S.R.); inge.nelissen@vito.be (I.N.); 3Institut Català de Nanociència i Nanotecnologia (ICN2), Consejo Superior de Investigaciones Científicas (CSIC), The Barcelona Institute of Science and Technology (BIST), Universitat Autònoma de Barcelona, Bellaterra, 08193 Barcelona, Spain; neus.bastus@icn2.cat (N.G.B.); victor.puntes@icn2.cat (V.F.P.); 4Centro Singular de Investigación en Química Biolóxica e Materiais Moleculares (CiQUS), Universidade de Santiago de Compostela, 15782 Santiago, Spain; beatriz.pelaz@usc.es; 5Grupo de Física de Coloides y Polímeros, Departamento de Química Inorgánica, Universidade de Santiago de Compostela, 15782 Santiago, Spain; 6Fachbereich Physik, CHyN, University of Hamburg, 22761 Hamburg, Germany; soliman0mahmoud@gmail.com (M.G.S.); wolfgang.parak@uni-hamburg.de (W.J.P.); 7Vall d’Hebron Research Institute (VHIR), 08035 Barcelona, Spain; 8Institució Catalana de Recerca i Estudis Avançats (ICREA), 08010 Barcelona, Spain

**Keywords:** inflammation, NLRP3, gold, nanorod, nanostar, nanosphere, cholesterol, oxidative phosphorylation, purinergic receptor, paraoxonase-2

## Abstract

The widespread and increasing use of engineered nanomaterials (ENM) increases the risk of human exposure, generating concern that ENM may provoke adverse health effects. In this respect, their physicochemical characteristics are critical. The immune system may respond to ENM through inflammatory reactions. The NLRP3 inflammasome responds to a wide range of ENM, and its activation is associated with various inflammatory diseases. Recently, anisotropic ENM have become of increasing interest, but knowledge of their effects on the immune system is still limited. The objective of the study was to compare the effects of gold ENM of different shapes on NLRP3 inflammasome activation and related signalling pathways. Differentiated THP-1 cells (wildtype, ASC- or NLRP3-deficient), were exposed to PEGylated gold nanorods, nanostars, and nanospheres, and, thus, also different surface chemistries, to assess NLRP3 inflammasome activation. Next, the exposed cells were subjected to gene expression analysis. Nanorods, but not nanostars or nanospheres, showed NLRP3 inflammasome activation. ASC- or NLRP3-deficient cells did not show this effect. Gene Set Enrichment Analysis revealed that gold nanorod-induced NLRP3 inflammasome activation was accompanied by downregulated sterol/cholesterol biosynthesis, oxidative phosphorylation, and purinergic receptor signalling. At the level of individual genes, downregulation of Paraoxonase-2, a protein that controls oxidative stress, was most notable. In conclusion, the shape and surface chemistry of gold nanoparticles determine NLRP3 inflammasome activation. Future studies should include particle uptake and intracellular localization.

## 1. Introduction

The use of engineered nanomaterials (ENM) is widespread and still increasing, as newly developed ENM enter the market frequently and in a large variety. Therefore, there is a great need to assess their effect on human health. While a significant volume of data exists on the role of chemical composition and size of ENM with respect to their hazards, there is a relative paucity of data on the role of ENM shape. Anisotropic ENM may display unexpected effects on cells and organisms, due, in part, on the altered distribution of surface energies and different adsorption of biocorona molecules at sites of high curvature that may exist on the surface of nanorods, -stars, -boxes, or -plates, as compared to nanospheres. However, often, such effects are not connected by the shape per se, as, in order to obtain different shapes, different synthesis routes have to be used, which often result in different surface chemistries. The typical example in this direction is gold nanorods, for which the standard synthesis requires a toxic surfactant, cetrimonium bromide (CTAB). Even if CTAB is removed after synthesis, residual CTAB may still be the reason for toxic effects. There are also synthesis routes for shape-controlled metal ENM that do not require synthetic surfactants, but are instead based only on biological molecules such as proteins [1]. However, the shape properties in such syntheses, in general, do not have the same quality as syntheses based on classical surfactants.

Anisotropic gold ENM have some attractive features for therapeutic, diagnostic, or theranostic use, as, upon modifying the shape, the optical properties may be tailored [2]. Gold nanorods are a promising platform for label-free, nanoplasmonic sensing in complex liquid samples, such as serum, based on their localized surface plasmon resonance peak, which is located in the near-infrared optical window and is highly sensitive to the surface binding of biomolecules [3]. Furthermore, in therapeutic applications involving intravenous administration, shape is of importance, as, e.g., in a microfluidic system that mimics the vasculature, gold nanorods show higher specific and lower nonspecific accumulation underflow at the target compared to their spherical counterparts [4]. In, e.g., photothermal therapy, gold nanoparticles are used in a large variety of shapes (shell, rod, cage, star, popcorn) for superior near-infrared photothermal efficacy and target selectivity [5]. The enhanced efficacy of anisotropic ENM shape in medical applications should, however, be balanced with a thorough safety evaluation. Indeed, NLRP3 inflammasome activation seems to be dependent on the shape of ENM [6], and this also holds for gold ENM [7]. However, again, for these studies, there is debate that the toxicity may actually also be connected to the particular surface coating used for this synthesis. In general, different physicochemical parameters of ENM are strongly entangled [8].

The NLR family, pyrin domain-containing 3 (NLRP3) inflammasome consists of a NLRP3 scaffold, an apoptosis-associated, speck-like protein containing a CARD (ASC) adaptor, and pro-caspase-1. Upon activation, NLRP3 recruits ASC. ASC then binds to pro-caspase-1, resulting in the autocleavage of this proenzyme to become the active enzyme, caspase-1. Caspase-1 processes pro-IL-1β and pro-IL-18 to bioactive IL-1β and IL-18, respectively [9]. These cytokines are potent mediators of inflammation. The NLRP3 inflammasome can be activated by various host-derived molecules, including extracellular ATP [10] and cholesterol [11]. Furthermore, mitochondria are involved in NLRP3 inflammasome activation [12]. Thus, NLRP3 inflammasome activation is closely linked to lipid and energy metabolism. Next to host-derived molecules and a multitude of infectious agents [9], the NLRP3 inflammasome can be induced by a wide range of xenobiotics, including ENM [13]. Its activation is associated with various inflammatory diseases, including lung fibrosis, obesity, and type-2 diabetes [14]. Moreover, its role is implicated in models for pancreatitis [15] and bronchiolitis obliterans syndrome [16].

Safety evaluation should preferably provide results quickly, since in that case, information on safety can be included in the choice of a specific ENM in the development process. Moreover, safety evaluation should preferably be amenable to high-throughput screening, given the large numbers of different ENM produced. An in vitro screening assay with demonstrated predictive value is suitable for this purpose. A further advantage of such an approach is that it reduces the use of animals, as animal experiments can be designed using knowledge obtained from in vitro experiments [17].

Our study aimed to evaluate whether gold nanorods, nanostars, and nanospheres were able to induce NLRP3 inflammasome activation and, if so, to identify the involved signal pathways. We showed here that gold nanorods, but not nanostars or nanospheres, could activate the NLRP3 inflammasome. Using microarrays, we showed that gold nanorod-induced NLRP3 inflammasome activation is accompanied by downregulated sterol/cholesterol biosynthesis, oxidative phosphorylation and purinergic receptor signalling. A role for oxidative stress is suggested.

## 2. Results

### 2.1. Particle Synthesis and Characterization

For this study, a set of six gold nanoparticles (GNPs) was synthesized. Details of their synthesis are reported in previous publications [18,19]. Electron microscopy (EM) pictures and the size distributions derived from these pictures are presented in Figure 1.

The synthesized GNPs comprised stars with 60 and 44 nm as an average of the longest tip-to-tip distance (GNP-1 and GNP-2, respectively), 30 and 50 nm spheres (GNP-3 and GNP-4, respectively), and rods of 40 × 16 nm and 60 × 14 nm (GNP-5 and GNP-6, respectively). The spherical and starshaped GNPs were synthesized in the presence of citric acid [19]. This weakly bound ligand was then exchanged to thiol-modified polyethylene glycol (PEG). In contrast, the rod-shaped GNPs had to be synthesized in the presence of CTAB, which was afterwards removed as well as possible by ligand exchange with thiol-modified PEG [18]. In this way, all the GNPs had, nominally, the same surface chemistry (e.g., PEG), though the presence of residual CTAB on the surface of the rod-shaped GNPs cannot be excluded. The suitability of all GNPs for biological applications was tested, ensuring solubility in water and stability in biologically relevant media [8]. Their characteristics are presented in Table 1.

### 2.2. Endotoxin Content of the Gold Nanoparticles

The absence of endotoxins is important for an accurate evaluation of GNP biological behaviour as endotoxin can be recognized by receptors, such as the toll-like receptor 4, which is abundant in the mononuclear phagocyte system, possibly interfering with immunological readouts, including NLRP3 inflammasome activation. All GNPs were synthesized in a laminar flow cabinet [20]. The endpoint chromogenic Limulus Amoebocyte Lysate (LAL) assay was used to measure the endotoxin content of 100 µg/mL GNP dispersions. The endotoxin threshold of contamination is 0.5 endotoxin units (EU)/mg (see, e.g., https://www.acciusa.com/pdfs/whitepapers/EndotoxinLimits_SeanJH-PR17-012.pdf (accessed on 23 March 2022)), so for a 100 µg/mL product concentration, this is 0.05 EU/mL. To establish possible interference in the assay by the GNPs, for each GNP, a similar GNP suspension was spiked with a 0.5 EU/mL endotoxin. For GNP-1 and -6, the endotoxin content was below the threshold of contamination of 0.05 EU/mL, and for GNP-2, it was slightly above this threshold, while interference precluded a correct endotoxin measurement for GNP-3, -4, and -5 (Table 2).

### 2.3. Inflammasome Activation

The development of the inflammasome activation assay used in the present study is described elsewhere [Vandebriel et al. submitted]. The assay uses THP-1 monocytes activated by phorbol myristate acetate (PMA). As a means of assay verification, we showed a concentration-dependent decrease in viability and concomitant increase in IL-1β and IL-18 production induced by 8–15 nm SiO_2_ NPs (NM202, JRC), an inducer of NLRP3 inflammasome activation [21]. This was substantiated by a strongly reduced response to these NP in cells deficient in the NLRP3 components ASC and NLRP3 [22]. Next, similar data were obtained using two types of aluminium adjuvant, another inducer of NLRP3 inflammasome activation [23,24]. The assay was further verified using the positive control nigericin.

In the first series of experiments, the PMA-activated THP-1 cells were exposed to a 2-fold dilution series (0, 3.12, 6.25, 12.5, 25, and 50 µg/mL; please note that all GNP concentrations refer to the elemental amount of Au, not taking into account the mass of the surface coating) of each of the six GNPs for 48 h (see Figure 2 for a representative experiment). For GNP-1 to GNP-5, depending on their specific stock concentration, a single concentration > 50 µg/mL was tested as well: 72, 82, 91, 76, and 100 µg/mL, respectively. Exposure to the gold nanostars, GNP-1 and GNP-2, as well as the gold nanospheres, GNP-3 and GNP-4, did not affect the viability and resulted in a small decrease in IL-1β and IL-18 production. In contrast, exposure to the gold nanorods, GNP-5 and GNP-6, resulted in a decreased viability and strongly increased IL-1β and IL-18 production. This may suggest that both nanorods activate the NLRP3 inflammasome, whereas both nanostars and both nanospheres do not. We note that the concentration range for the reduction of the viability upon exposure to rod-shaped GNPs versus spherical GNPs is compatible with previous results obtained with similar PEGylated GNPs [8].

In the second series of experiments we selected, based on Figure 2 and Table 2, one of each of the gold shapes, namely, the gold nanostars, GNP-1, the gold nanospheres, GNP-4, and the gold nanorods, GNP-6. For each of the GNP shapes, the differences in the dose–response curves between the two GNPs were only minor, the most notable difference being a stronger IL-1β and IL-18 induction by GNP-6 compared to GNP-5. The endotoxin content of GNP-1 and GNP-6 was below the detection limit (A in Table 2), making them the GNPs of choice for nanospheres and nanorods, respectively, while a choice based on endotoxin content between GNP-3 and GNP-4 could not be made. At any rate, relating endotoxin content (or lack of information thereof) and cytokine levels suggested that enhanced cytokine levels were not due to endotoxin stimulation.

Each of the GNPs was tested in three independent experiments (see Figure 3 for a representative experiment). Averaging these three independent experiments showed that exposure to GNP-1 nanostars at a concentration of 28.6 µg/mL (2^4.84^ µg/mL) did not significantly affect the viability (104 ± 6% of the control; mean ± SD) and resulted in a 50% decrease in IL-1β production (50 ± 7% of the control). Exposure to GNP-4 nanospheres at a concentration of 24.7 µg/mL (2^4.63^ µg/mL) resulted in a 50% decrease in viability (52 ± 10% of the control) and had no significant effect on IL-1β production (108 ± 76% of the control). Exposure to GNP-6 nanorods at a concentration of 33.1 µg/mL (2^5.05^ µg/mL) resulted in a small decrease in viability (83 ± 3% of the control) and a 50% increase in IL-1β production (151 ± 101% of the control). The GNP-6 nanorods were also tested at much higher weight concentrations than the GNP-1 nanostars and the GNP-4 nanospheres. In this second experiment, we wished to express the effects relative to two different dose metrics, particle (Au) weight and particle number. Relative to the particle number, GNP-6 nanorods have a 10-fold higher weight (1.324 µg/mL corresponding to 1 nM) compared to GNP-1 nanostars (0.142 µg/mL) and GNP-4 nanospheres (0.123 µg/mL). The particle number concentrations (in nM) were estimated based on assumptions of molar mass for each GNP made by TEM imaging, which is, however, prone to errors, as, in particular, for the starshaped Au NPs, their volume and, thus, the number of gold atoms per Au NPs could not be precisely determined. In the Appendix A, the same data as Figure 3 are presented, but using particle number concentration (nM) as dose metric. Exposure at 133 µg/mL (2^7.05^ µg/mL) resulted in a strong decrease in viability (30 ± 5% of the control) and an almost 6-fold increase in IL-1β production (569 ± 363% of the control). At higher concentrations, viability dropped to close to zero, while the IL-1β produced remained at a rather similar level.

To confirm that the effects on viability and IL-1β production were related to NLRP3 inflammasome activation, the response of PMA-treated, wildtype THP-1 cells was compared to the response of similarly treated ASC- and NLRP3-deficient THP-1 cells. These ASC- and NLRP3-deficiencies did not affect viability, since for GNP-1 nanostars at a concentration of 28.6 µg/mL, the viability of ASC-deficient cells was 103 ± 5%, of NLRP3-deficient cells, 114 ± 8%, and of wildtype cells, 104 ± 6%. For GNP-4 nanospheres at a concentration of 24.7 µg/mL, viability of ASC-deficient cells was 50 ± 18%, of NLRP3-deficient cells, 55 ± 12%, and of wildtype cells, 52 ± 10%. For GNP-6 nanorods at a concentration of 33.1 µg/mL, viability of ASC-deficient was 90 ± 13%, of NLRP3-deficient cells, 81 ± 0%, and of wildtype cells, 83 ± 3%. No IL-1β production was seen in either type of deficient cells at the indicated concentrations, regardless of the type of GNP.

At high concentrations (530 and 1060 µg/mL) of GNP-6 nanorods, some IL-1β production (9.9% and 7.1%, respectively, of the wildtype cells) was induced in the NLRP3-deficient cells, whereas IL-1β was not induced in the ASC-deficient cells. Residual background in the NLRP3-deficient cell line may be explained by only partial silencing of the *NLRP3* gene by the short hairpin RNA. Similar “leakiness” of *NLRP3* gene knockdown was observed after exposure to alum (Al(OH)_3_ and AlPO_4_) adjuvants [Vandebriel et al., submitted]. Nonetheless, the results do show a clear difference between the responses of the wildtype and deficient cells.

We also analysed NLRP3 inflammasome activation by dose–response modelling. The three replicate experiments, of which a representative one is presented in Figure 3, were analysed together. Dose–response modelling using the software program PROAST [25] was used to generate concentrations that induced a 10% reduced viability (Effective Dose (ED)_10_), including a 90% Confidence Interval (CI90)). An ED_10_ and CI90 could be calculated for gold nanospheres and nanorods (Table 3). This was not possible for gold nanostars due to a lack of dose–response relationship. In ASC- and NLRP3-deficient cells, the ED_10_ values were 30% and 90% higher, respectively, for nanospheres, while they were 5-fold higher for nanorods. This suggests that the effect of ASC- or NLRP3-deficiency is more profound for nanorods than for nanospheres. For IL-1β production, an ED_10_ could not be calculated due to a lack of a dose–response relationship. In conclusion, our results suggest a more important role for NLRP3 inflammasome activation in reducing viability for nanorods compared to nanospheres.

### 2.4. Microarray Analysis

To study the effects of the three GNPs on gene expression, PMA-activated THP-1 cells were exposed to GNP-1 nanostars, GNP-4 nanospheres, and GNP-6 nanorods for 48 h. Next, to study the role of the NLRP3 inflammasome in the effects on gene expression, ASC-deficient and NLRP3-deficient THP-1 cells were exposed to GNP-6 nanorods for the same time period. Based on the data presented in Appendix A, we chose an estimated particle number-based concentration of 0.05 nM, as this was the highest concentration not inducing significant cytotoxicity (71% viability for GNP-6 exposed wildtype THP-1 cells; 79% or higher for the other GNP-cell line combinations). Three independent exposure experiments were performed. The average cell viability is shown in Figure 4.

Similar to Appendix A, THP-1 cells exposed to 0.05 nM GNP-6 nanorods showed a viability of 71%.

RNA was isolated from exposed cells of three biological replicate experiments, subjected to QC and analysed using microarrays. The RNA Integrity Number of all samples was 8.3 or higher. Gene expression analysis identified 103 differentially expressed genes (DEGs; Table 4). Their expression was significantly affected by at least one exposure, as compared to unexposed control cells (*p* < 0.001). In wildtype THP-1-derived cells, nanorods induced more DEGs (15) than nanostars (6) or nanospheres (9). In wildtype, THP-1-derived cells, nanorods induced fewer DEGs (15) than in ASC-deficient (35) or NLRP3-deficient (40) THP-1-derived cells. Comparison between unexposed cells derived from wildtype vs. ASC-deficient, THP-1-derived cells revealed 332 DEGs, while comparison between unexposed cells derived from wildtype vs. NLRP3-deficient, THP-1-derived cells revealed 219 DEGs.

Principal component analysis (PCA) is a mathematical means of dimensionality reduction used in order to visualize the differences in gene expression between samples. The DEGs from both panel A and panel B of Table 3 were used as input. The PCA (Figure 5) shows three clusters, coinciding with each of the THP-1-derived cell models: (i) The cluster on the left comprises the wildtype cells, with some distinction between the GNP-6 nanorod-exposed group (red), and the unexposed control group (black). In addition, some distinction between the GNP-1 (green) nanostar and GNP-4 (blue) nanosphere-exposed groups and the unexposed control group is seen, without a distinction between the GNP-1 and GNP-4-exposed groups. (ii) The cluster on the bottom right comprises the ASC-deficient cells, with a clear distinction between the GNP-6 nanorod-exposed group (red) and the unexposed control (black). (iii) The cluster on the top right comprises the NLRP3-deficient cells, with some distinction between the GNP-6 nanorod-exposed group (red) and the unexposed control (black).

Figure 6 shows the heatmap of the 103 DEGs for GNP treatments vs. unexposed controls in wildtype, ASC-deficient, and NLRP3-deficient THP-1-derived cells (see panel A in Table 4). It shows that the differential gene expression responses to the GNP-6 nanorods were (1) rather different compared to those to the GNP-1 nanostars and GNP-4 nanospheres, and (2) rather comparable in the three cell types (WT, ASC-deficient, and NLRP3-deficient).

In order to analyse the responses at a functional or pathway level, we next performed Gene Set Enrichment Analysis (GSEA; Figure 7). The overall picture is a clear separation of affected pathways between GNP-1 nanostars and GNP-4 nanospheres on the one hand, and GNP-6 nanorods on the other hand. This is in line with the viability and cytokine data (Figure 2 and Figure 3), the PCA plot of DEGs (Figure 5), and the heatmap (Figure 6), which also showed a clear division between nanostars and nanospheres vs. nanorods. The pathways upregulated by nanostars and nanospheres are all involved in the cell cycle. Several pathways are downregulated by nanorods, including cholesterol synthesis, lipid storage and transport, oxidative phosphorylation/respiratory chain/ATP synthesis, and purinergic signalling (Figure 7).

Most pathways affected by the nanorods can be divided into two groups, one group involved in lipid metabolism and the other one in energy metabolism. It should be mentioned that lipid metabolism and energy metabolism are closely interrelated [26].

At the level of individual genes, among the genes most notably downregulated by nanorods in wildtype, ASC-deficient and NLRP3-deficient cells are methylsterol monooxygenase 1 (MSMO1), cytochrome P450 family 51 subfamily A member 1 (CYP51A1), and mevalonate kinase (MVK), all of which are involved in cholesterol synthesis (Figure 8). Additionally, the mitochondrial genes mitospecific ribosomal protein (MRP) L17 (MRPL17), MRPL30, G elongation factor mitochondrial 1 (GFM1), and TP53-induced glycolysis and apoptosis regulator (TIGAR) are present in this set. Furthermore, a strong downregulation by nanorods of Paraoxonase-2 (PON2) is found.

## 3. Discussion

Toxicity of GNPs is related to surface catalytic effects. The larger the surface-to-volume (which is equivalent to the surface-to-mass for the same elemental composition) ratio, the higher, in general, the reduction in cell viability [8]. In order to find out if the effect of the nanorods in Figure 2 is merely due to a bigger surface area at the same exposure concentration, we can compare the surface areas of the different GNPs. Considering the rod-shaped GNP-6 as a cylinder with flat caps (which is a rough estimation), each GNP has a surface-area-to-volume ratio of [2π·r·(r + h)]/[π·r^2^·h] = 2·(r + h)/(r·h) = 2·(14/2 nm + 60 nm)/(14/2 nm·60 nm) ≈ 0.32 nm^−1^. It has to be noted that this is a rough estimate given the size- and shape-distribution of the sample. On the other hand, one spherical shaped GNP-3 has a surface area to volume ratio of [4π·r^2^]/[(4/3) π·r^3^] = 3/r = 3/(30/2 nm) ≈ 0.2 nm^−1^. At the same mass concentration (i.e., the same volume of Au NPs), the GNP-6 sample has a 0.32 nm^−1^/0.2 nm^−1^ ≈ 1.6 higher surface area than the GNP-3 sample. Similar discussions can be found in a previous report [27]. Therefore, the effect of the rod-shaped GNP samples, as shown in Figure 2, is not due to differences in surface-to-volume ratio. We note that, due to their complex shape, we were not able to give a reliable value for surface and volume of the starshaped Au NPs.

The pathways upregulated by the nanostars and nanospheres are all involved in the cell cycle. This may suggest that these two shapes of GNP affect cell proliferation. This is not readily reflected in the in vitro data, as little or no effect on cell viability is seen at the concentrations tested. The GO term “receptor internalization” was the only one downregulated by the nanostars and nanospheres.

As shown in Figure 6, the differential gene expression responses to the GNP-6 nanorods were rather comparable in the three cell types (WT, ASC-deficient, and NLRP3-deficient). This may suggest that most of the affected pathways are upstream of the NLRP3 inflammasome. It should be noted that NLRP3 inflammasome activation results in protein cleavage (of pro-caspase-1, pro-IL-1β, pro-IL-18, and gasdermin D) do not necessarily affect the expression of these genes. Apparently, downstream events, e.g., pyroptosis, are not accompanied by major effects on gene expression at the GNP concentration tested.

One of the pathways downregulated is cholesterol synthesis. Cholesterol has been shown to repress NLRP3 inflammasome activation by antagonizing sterol response element–binding protein [28]. As such, it is possible that reduced expression of the genes involved in sterol/cholesterol synthesis may be related to NLRP3 inflammasome activation by gold nanorods. More recent studies have shown a relation between cholesterol signalling and NLRP3 inflammasome activation [29] and between cholesterol trafficking and NLRP3 inflammasome activation [30].

Generally, M1 macrophages have been associated with glycolysis, allowing for rapid ATP production in acute inflammation or antibacterial defence, whereas M2 macrophages have been linked to oxidative phosphorylation to occur in more energy efficient, long-term resolution and repair. From this notion, it may be suggested that downregulation of oxidative phosphorylation can be linked to upregulation of glycolysis. Activation of glycolysis is linked to NLRP3 inflammasome activation [31]. It is possible that the negative regulation of lipid storage and transport may result in a higher lipid content, in line with NLRP3 inflammasome activation [32]. While PMA-activated THP-1 cells are regarded as M0 cells, downregulation of oxidative phosphorylation due to gold nanorod exposure may still be linked to NLRP3 inflammasome activation.

Purinergic signalling is mediated by purine nucleotides and nucleosides, such as adenosine and ATP. The binding of ATP to the P2X7 purinergic receptor results in NLRP3 inflammasome activation [33]. This process has been previously described for SiO_2_ and TiO_2_ nanoparticles [34,35,36]. Thus, the observation of downregulated purinergic signalling may contrast NLRP3 inflammasome activation.

CTAB is used in the synthesis of gold nanorods, and plays a role in their potential cytotoxicity. Removal of excess CTAB from a dispersion of CTAB-stabilized poly(ethylene glycol) (PEG)-modified gold nanorods strongly reduced their cytotoxicity [37]. CTAB can enter cells with or without gold nanorods and, in the cells, damage mitochondria and induce apoptosis [38]. No apoptotic or mitochondrial damage pathways are significant in the GSEA. In conclusion, the data do not suggest a dominant role of CTAB in the observed effects of the gold nanorods.

An aspect not addressed in our study is a possible difference in cellular uptake between the different shapes of PEGylated GNPs. The uptake of 45 × 10 nm PEGylated gold nanorods by murine RAW264.7 macrophages was lower than that of 50 nm PEGylated nanospheres [39], while the uptake of 91 nm PEGylated gold nanorods by these cells was higher than that of 83 nm PEGylated gold nanostars [40]. In contrast, studies with similar PEGylated GNPs as those used in this study suggest, for some cell lines, a higher uptake of nanorods than of nanospheres (with comparable geometries to GNP-6 and GNP-4, respectively), though a strong influence of the cell line used was noted [8]. Data are thus insufficient to relate the biological effect of the gold nanorods observed here to differences in their cellular uptake quantity.

The study did not investigate the mechanisms underlying the cell death observed. Most importantly, a possible role of apoptosis could have been evaluated by measuring caspase-3 activity.

In a model of reactive oxygen species (ROS) production induced in Ea.hy926 endothelial cells by 2,3-dimethoxy-1,4-naphthoquinon, *PON2* gene knockdown increased ROS production [41]. Furthermore, in a model of ER stress-induced ROS production, induced in vascular smooth muscle cells by TGF-β1, *PON2* gene knockdown resulted in increased ROS production [41]. In conclusion, reduced *PON2* expression may result in increased ROS production. It is possible that this increased ROS production may in turn induce NLRP3 inflammasome activation [42]. Next, *PON2* induces a switch in macrophage polarization towards M2 [43]. It is possible that reduced *PON2* gene expression is in line with M1 macrophage polarization, as deduced from downregulated oxidative phosphorylation, as described above.

In conclusion, PEGylated gold nanorods, but not nanostars or nanospheres, showed NLRP3 inflammasome activation in wildtype, THP-1-derived cells, while cells deficient in the ASC adaptor or NLRP3 scaffold did not show this effect. The pathways related to gold nanorod-induced NLRP3 inflammasome activation are downregulated sterol/cholesterol biosynthesis, oxidative phosphorylation, and purinergic receptor-signalling. At the level of individual genes, the most notable finding was a strong downregulation of PON2, which is possibly related to increased ROS production. We have shown that the shape and surface chemistry of gold nanoparticles determine NLRP3 inflammasome activation. Future studies should include particle uptake and intracellular localization.

## 4. Materials and Methods

### 4.1. Particle Synthesis and Characterization

#### 4.1.1. General Information

All chemicals were obtained from Sigma-Aldrich, except PEG ligand (HS−C11−EG6−OCH2−COOH), which was obtained from Rapp Polymere. All the stock solutions of the reagents, except for PEG ligand (HS−C11−EG6−OCH2−COOH), were filtered through a 0.2 μm Millipore syringe filter prior to use. All plasticware used was endotoxin-free, and all glassware used was previously cleaned with aqua regia and thoroughly rinsed with endotoxin-free water.

#### 4.1.2. Synthesis of GNPs

Preparation was carried out in aqueous solution by reduction of a gold precursor. In the case of rod-shaped GNPs, surfactant was used for shape control. After the preparation, a ligand exchange reaction was used in order to normalize the surface of all particles with carboxy-PEG thiol ligand, providing the particles with the same surface chemistry. All samples were characterized by transmission electron microscopy (TEM; FEI Tecnai G2 20 TWIN, Thermo Fisher Scientific, Hillsboro, OR, USA), UV−vis absorption spectroscopy, and elemental analysis for determining their concentration. For a detailed description about the synthesis and characterization of the GNPs, we refer to previous publications [8,19].

### 4.2. Inflammasome Activation

#### 4.2.1. NP Dispersion and Exposure

The GNPs were sonicated in a sonication bath (Branson 1800) for 1 min and vortexed for 1 min to redisperse any GNP that might have aggregated/agglomerated. To obtain the desired exposure concentrations, a 2-fold dilution series of the GNPs was prepared in sterile water (Fluka, Thermo Fischer Scientific, Landsmeer, the Netherlands). Each GNP dilution was then diluted 10 times in complete cell culture medium (CCM), that is, RPMI 1640 (Gibco, Thermo Fisher Scientific, Landsmeer, the Netherlands) supplemented with foetal calf serum (10% *v*/*v*, Greiner-Bio), penicillin (100 U/mL), and streptomycin (100 µg/mL) (Gibco). The GNPs in CCM were added to the same volume of PMA-activated THP-1 cells (see below).

Due to a difference in the concentration of the various GNP dispersions supplied, in the first series of experiments, the highest exposure concentration was 72 µg/mL (GNP-1), 82 µg/mL (GNP-2), 91 µg/mL (GNP-3), 76 µg/mL (GNP-4), 100 µg/mL (GNP-5), or 50 µg/mL (GNP-6). For each of the GNPs, the dilution series included 50, 25, 12.5, 6.25, and 3.125 µg/mL.

In the second series of experiments, for GNP-1 and GNP-4, the dilution series was 0.2, 0.1, 0.05, 0.025, 0.0125, 0.00625, and 0.003125 nM, while for GNP-6, the dilution series was 0.8, 0.4, 0.2, 0.1, 0.05, 0.025, and 0.0125 nM. This is equivalent to a dilution series for GNP-1 of 28.6, 14.3, 7.2, 3.6, 1.8, and 0.9 µg/mL; for GNP-4 24.7, 12.3, 6.2, 3.1, 1.5, and 0.8 µg/mL; and for GNP-6 1060, 530, 265, 132.5, 66.3, and 33.1 µg/mL.

#### 4.2.2. Cell Line Maintenance

THP-1 cells (ATCC TIB-202), THP1-defASC cells, and THP1-defNLRP3 cells (InvivoGen, Toulouse, France) were used in this study. The cells were cultured in CCM. Additionally, the ASC- and NLRP3-deficient THP-1 cells were cultured in CCM supplemented with HygroGold (200 µg/mL, InvivoGen) to maintain the siRNA responsible for suppression of ASC or NLRP3. All cell lines were sub-cultured twice per week, seeded to a cell density of 2 × 10^5^ cells/mL, and not allowed to grow to a density greater than 1 × 10^6^ cells/mL. Cells were not cultured beyond twenty passages to prevent genetic divergence.

#### 4.2.3. Maturation of THP-1 Cells

The wildtype, ASC-deficient, and NLRP3-deficient THP-1 cells were differentiated into macrophage-like cells by culturing for 3 h in the presence of 100 ng/mL phorbol 12-myristate 13-acetate (PMA) (Sigma, Merck Life Science, Amsterdam, the Netherlands) in 96-well format at a cell density of 5 × 10^5^ cells/mL, 100 µL/well. After this incubation, the cells were adherent. The medium was replaced with fresh culture medium without PMA and the plates were incubated for 24 h at standard conditions (humidified incubator at 37 °C, 5% CO_2_). After this incubation period, the cells were exposed to a two-fold dilution series of GNPs 1-6, as described above, for 48 h at standard conditions. Cells were used for viability testing or RNA isolation; culture supernatants were frozen at −80 °C until further use (ELISA).

#### 4.2.4. Viability

The viability of the cells after exposure was assessed using the cell proliferation reagent WST-1 (Sigma-Aldrich). Exposed cells (and controls) were incubated for 2.5 h under standard conditions in the presence of 10% (*v*/*v*) WST-1 reagent. After incubation, in each well, the absorbance (A) was measured at 440 nm (A440) and corrected for background absorbance at 620 nm (A620). Exposures for viability assessment were performed in triplicate and the viability was calculated as follows: (A (cells in medium, X)—A (medium only, X))/A (cells in medium, C)—A (medium only, C), where X is a specific concentration GNPs and C the control. The viability was expressed as percentage of the control.

As a control, for each GNP at the highest exposure concentration (in medium), the A440–A620 signal was measured and was confirmed not to interfere with the readout signal of the WST-1 assay.

#### 4.2.5. IL-1β and IL-18 ELISA

The IL-1β and IL-18 concentrations in the culture supernatant were determined using ELISA kits (eBioscience, Thermo Fisher, Landsmeer, the Netherlands) according to the manufacturer’s instructions. An 8-point, 2-fold dilution series of a cytokine standard was prepared; diluent was used as blank. A calibration curve was calculated using 5-parameter curve fitting. Exposures for the assessment of IL-1β and IL-18 secretion were performed in four wells per condition. Culture supernatants were frozen at −80 °C until further use. In the case of IL-1β, the supernatants were tested in twenty-fold dilution to stay within the standard curve concentration range of the ELISA kit.

#### 4.2.6. Cell Harvest for RNA Isolation

After centrifugation of the plates and taking off the supernatants, 120 µL consisting of 5 parts RNAprotect Cell Reagent (Qiagen, 76526, Venlo, The Netherlands) and 1 part PBS (Gibco, 20012-019, Thermo Fisher Scientific, Landsmeer, the Netherlands) was added to each well. The cell lysates were frozen at −80 °C until further use.

#### 4.2.7. Dose–Response Modelling

Dose–response modelling for viability and IL-1β production was performed with the statistical software package PROAST [25] within the software environment ‘R’ [44].

In this approach, a dose–response dataset is evaluated as a whole by fitting a dose–response model over the entire dose range studied. Having fitted a dose–response model to the data, this curve is used to assess the benchmark dose (BMD) associated with the benchmark response (BMR) of 50%. The choice of the model for deriving the BMD follows from a procedure of applying likelihood ratio tests to the five members of the following two nested families of models:

Exponential family
E1: y = a
E2: y = a exp(b x)
E3: y = a exp(b x^d^)
E4: y = a (c − (c − 1)) exp(b x)
E5: y = a (c − (c − 1)) exp(b x^d^)

Hill family
H1: y = a
H2: y = a (1 − x/(b + x))
H3: y = a (1 − x^d^/(bd + x^d^))
H4: y = a (1 + (c − 1) x/(b + x))
H5: y = a (1 + (c − 1) x^d^/(bd + x^d^))
where y is any continuous endpoint and x denotes the dose. In these models, the parameter a represents the background response and the parameter b can be considered the parameter reflecting the efficacy of the chemical (or the sensitivity of the subject). First, the likelihood ratio test was used to establish whether extension of a model by increasing the number of parameters resulted in a statistically significant improvement of the fit. The model that could not be significantly improved was considered the most appropriate member (which adequately fits but does not overfit the data) within each family. In addition, a goodness of fit test (*p* > 0.05) was applied by comparing the log-likelihood of the fitted model to that associated with the so-called “full model.” The full model simply consists of the observed (mean) responses at each applied dose. The model is accepted when the log-likelihood value of the fitted model is not significantly worse than that of the full model. Subsequently, the BMDs are derived from the different models and the 90% confidence intervals (CIs) surrounding the BMDs are calculated using the profile-likelihood method. The BMD used in the analysis was the geometric average of the BMDs derived for the different models. The 90% CI surrounding this BMD comprised the BMDL and the BMDU found for the BMD estimates derived from the different models.

### 4.3. Microarray Analysis

#### 4.3.1. RNA Isolation

For each exposure condition, the cell lysates of two replicate wells were pooled to obtain sufficient RNA. Total RNA was isolated using the miniRNeasy RNA isolation kit, according to the specifications of Qiagen. The RNA concentration was determined using a NanoDrop Spectrophotometer (NanoDrop Technologies, Wilmington, DE, USA). RNA integrity was analysed using the Agilent 2100 Bioanalyzer (Agilent Technologies, Diegem, Belgium). Samples showing an RNA Integrity Number value above 7 were qualified for microarray analysis. RNA was stored in RNase-free water (Qiagen, Venlo, The Netherlands) at −80 °C until further use.

#### 4.3.2. RNA Amplification and Labelling

Labelled RNA was prepared and hybridized for each of the eight test conditions. This was performed for three biological replicates, resulting in 24 samples. Fluorescently labelled samples were obtained using the Quick Amp Labelling kit (Agilent Technologies, Amstelveen, the Netherlands). Briefly, 1 µg of total RNA was reverse transcribed into complementary DNA (cDNA) using a T7-promotor primer and MMLV reverse transcriptase. The cDNA was transcribed into complementary RNA (cRNA), during which, it was fluorescently labelled by incorporation of Cy3-CTP. After purification using the RNeasy mini kit (Qiagen), cRNA yield and specific activity were determined using a NanoDrop spectrophotometer (NanoDrop Technologies, Wilmington, DE, USA).

#### 4.3.3. Microarray Hybridization and Raw Data Processing

Equal amounts of Cy3-labelled samples were hybridized onto Whole Human Genome 4 × 44 K one-color oligonucleotide arrays (G4112F, Agilent Technologies) for 17 h in a Tecan HS 4800TM Pro hybridization Station (Tecan Benelux BVBA, Mechelen, Belgium). The arrays were scanned on an Agilent G2565BA microarray scanner and further processed using Agilent Feature Extraction Software (version 10.7.3.1). For each feature (or spot) on the array, the feature extraction output file contains the Cy3 fluorescence signal, together with feature quality and gene information. Fluorescence signals were log2-transformed, quantile normalized, and filtered with regard to reliability of the feature. Reliable features were defined as features: (1) of which the intensity was higher than the feature noise level, (2) of which the intensity was below the feature saturation level, (3) that were not a population outlier, (4) that were uniform spots, and (5) that were not a control feature. Finally, data were collapsed by gene symbol, resulting in a table for 10,246 genes and 24 samples.

#### 4.3.4. Data Analysis

Data were further analysed within the software environment ‘R’ (Version 3.4.3). For each exposure, gene expression data were compared to their control group by means of a *t*-test, and genes significant at *p* < 0.001 in at least one comparison were selected. Expression data were visualized as a heatmap combined with hierarchical clustering.

Pathway-level analysis was performed using Gene Set Enrichment Analysis (GSEA) [45].

We compared gold nanostar-exposed to nonexposed wildtype cells, gold nanosphere-exposed to nonexposed wildtype cells, gold nanorod-exposed to nonexposed wildtype cells, gold nanorod-exposed to nonexposed ASC-deficient cells, and gold nanorod-exposed to nonexposed NLRP3-deficient cells. Additionally, we compared nonexposed, ASC-deficient and NLRP3-deficient cells to nonexposed wildtype cells.

## Figures and Tables

**Figure 1 ijms-23-05763-f001:**
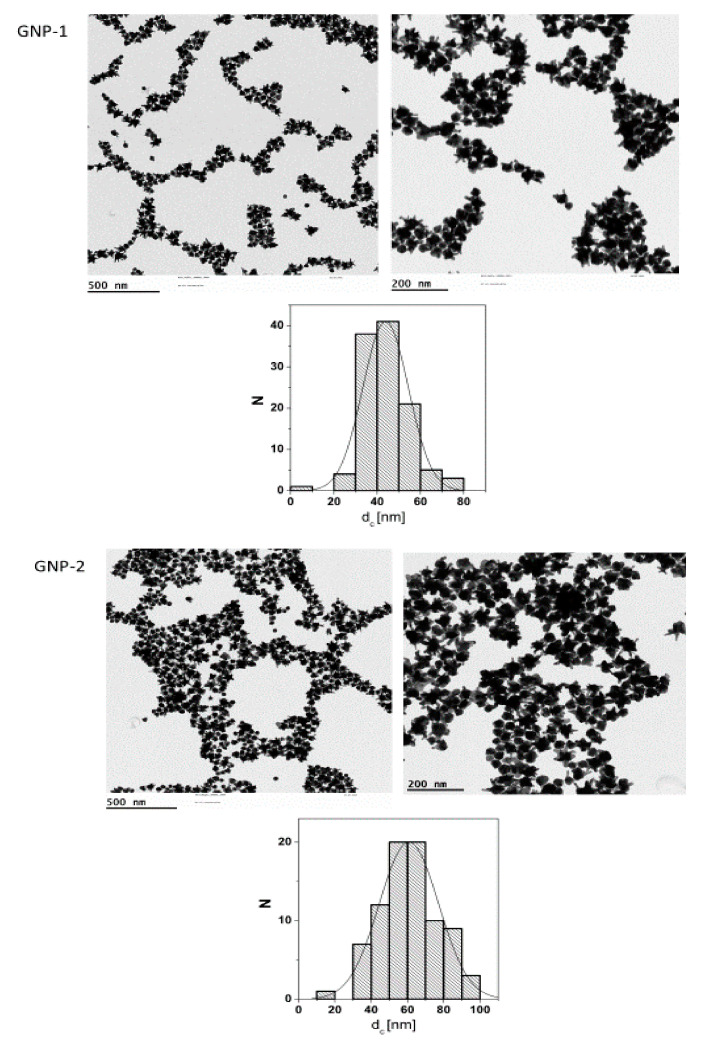
Electron microscopy (EM) pictures of gold nanoparticles (GNP) 1-6 and the size distributions derived from them. D, diameter; L, length; N, number.

**Figure 2 ijms-23-05763-f002:**
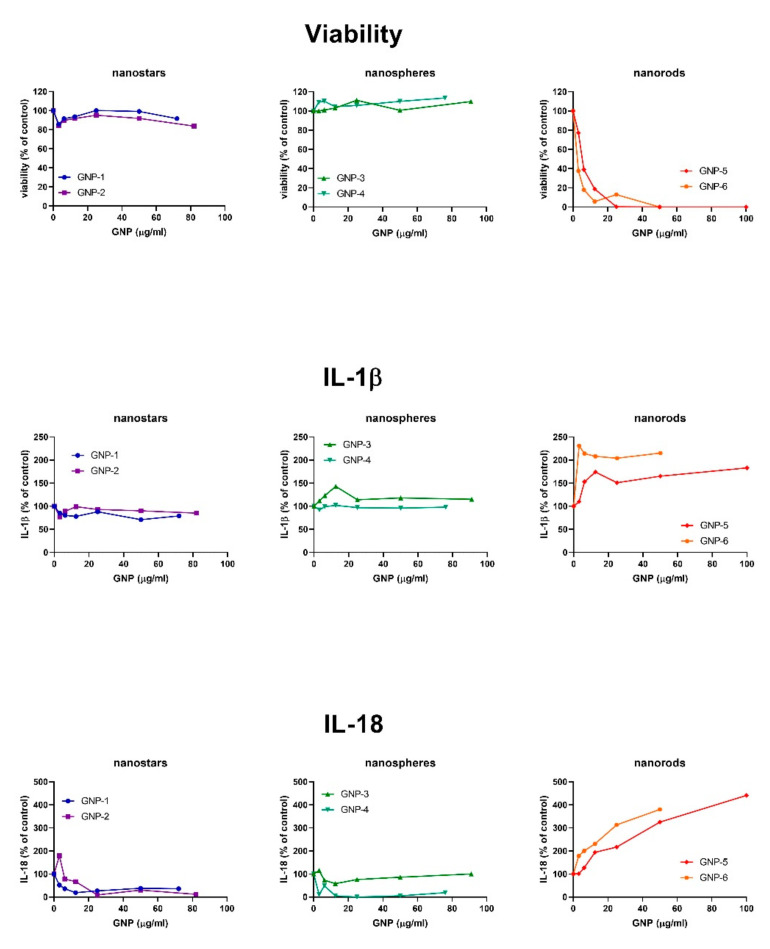
Exposure effects of gold nanostars (GNP-1, GNP-2), nanospheres (GNP-3, GNP-4), and nanorods (GNP-5, GNP-6) on cell viability and IL-1β and IL-18 production of THP-1-derived cells. THP-1 cells were PMA-stimulated for 3 h, allowed to rest for 24 h, and exposed for 48 h, after which, cell viability and cytokine production were measured. Percentage of untreated controls. Representative result of *N* = 2 (nanostars, nanospheres) or *N* = 3 (nanorods) independent experiments.

**Figure 3 ijms-23-05763-f003:**
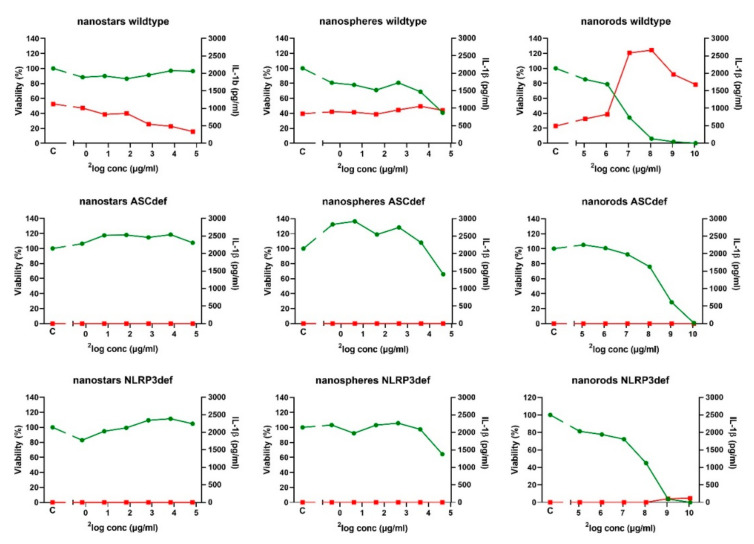
Exposure effects of gold nanostars (GNP-1), nanospheres (GNP-4), and nanorods (GNP-6) on cell viability (green) and IL-1β production (red) of cells derived from wildtype, ASC-deficient and NLRP3-deficient THP-1 cells. Cells were PMA-stimulated for 3 h, allowed to rest for 24 h, and exposed for 48 h, after which, cell viability and cytokine production were measured. Nanoparticle concentration is expressed as ^2^log (µg/mL). Viability: percentage of untreated controls. IL-1β: concentration in pg/mL. Representative result of *N* = 3 independent experiments. C: untreated control.

**Figure 4 ijms-23-05763-f004:**
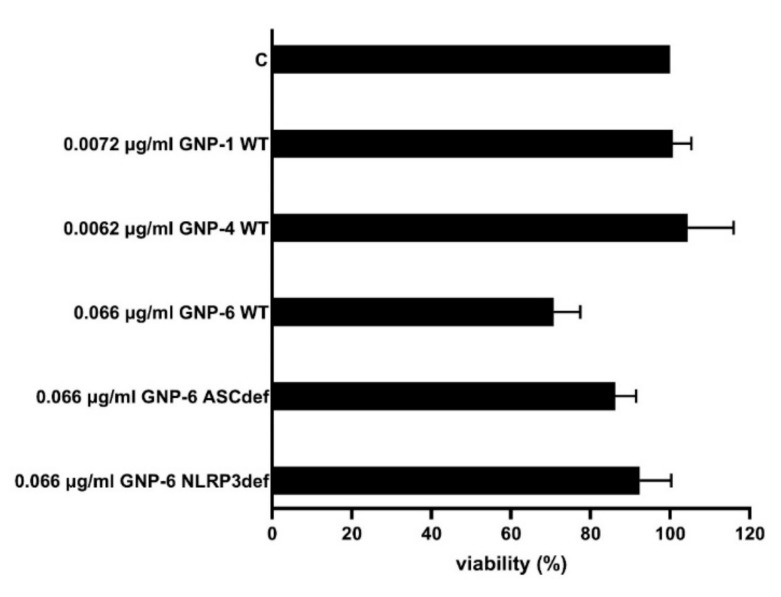
Viability of cells used for gene expression analysis. PMA-activated THP-1 wildtype (WT) cells were exposed to GNP-1, GNP-4, and GNP-6, and PMA-activated, ASC-deficient, and NLRP3-deficient cells were exposed to GNP-6. Cells were PMA-stimulated for 3 h, allowed to rest for 24 h, and exposed for 48 h, after which, cell viability was measured. Viability: percentage of untreated controls ± SD (*N* = 3). All exposure concentrations were 0.05 nM; this translates to the different weight concentrations indicated.

**Figure 5 ijms-23-05763-f005:**
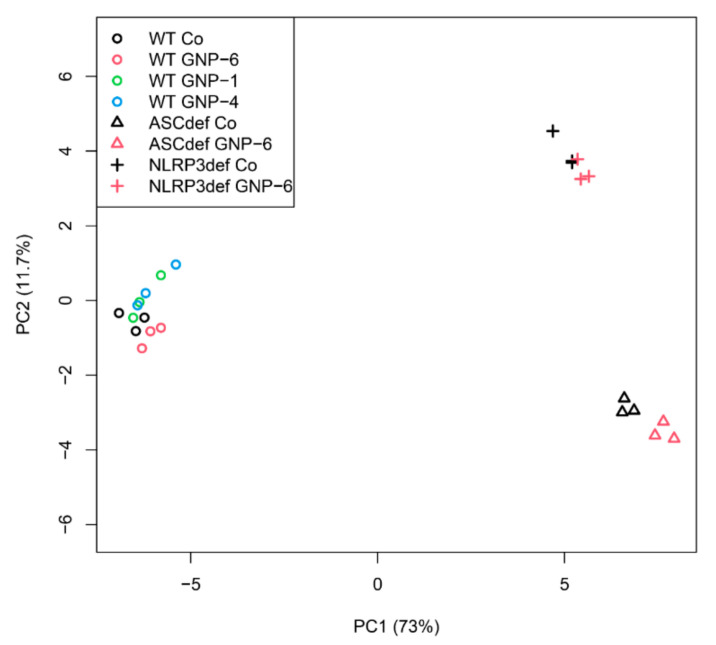
Principle Component Analysis (PCA) plot of DEGs. The axes represent the dimensions of the largest (principal component 1 (PC1); in this case 73%) and the second largest (PC2; in this case, 11.7%) variation between the samples. All DEGs from Table 3 were used as input data. The numbers at the X- and Y-axes are arbitrary units. Co: unexposed controls.

**Figure 6 ijms-23-05763-f006:**
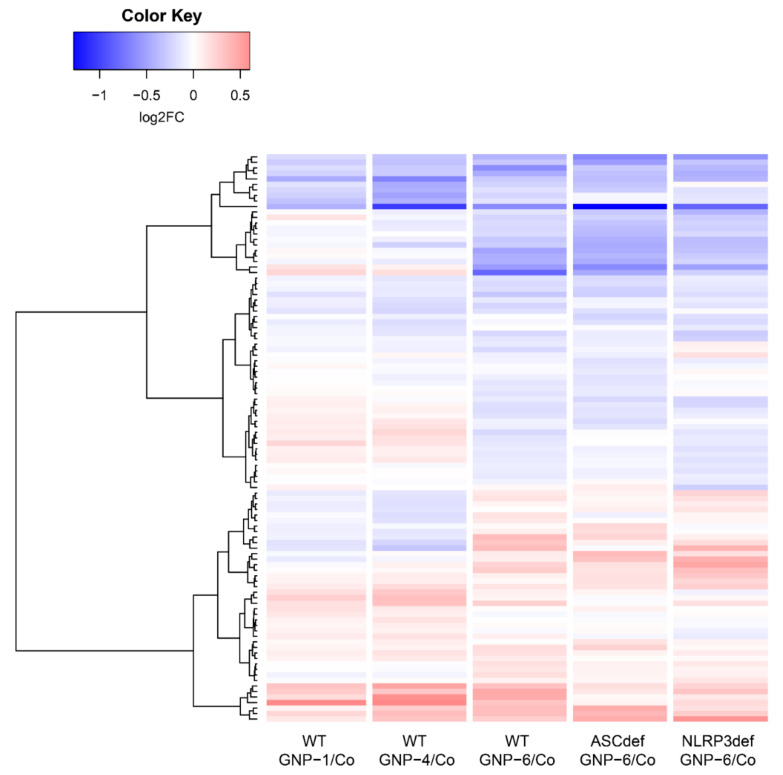
Heatmap of the 103 DEGs showing average responses per exposure group compared to their respective controls. Co: unexposed controls; log2FC: ^2^log of the fold change (FC).

**Figure 7 ijms-23-05763-f007:**
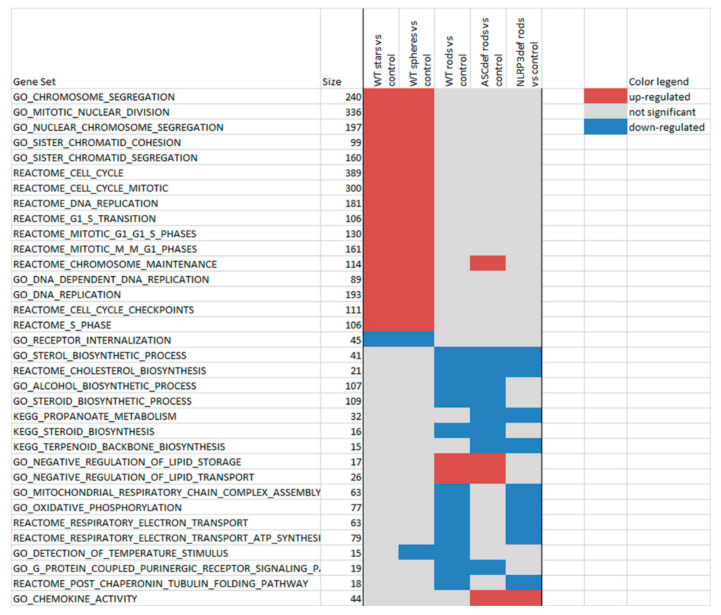
Gene Set Enrichment Analysis. The pathway selection is based on pathways that are significant (*p* < 0.001) multiple times, as well as in the top 20 per exposure multiple times. Red: upregulated; blue, downregulated; no colour: not significantly altered. Size: number of genes on the microarray (for a specific pathway). The pathways were ordered (1) based on the response and (2) alphabetically within each type of response.

**Figure 8 ijms-23-05763-f008:**
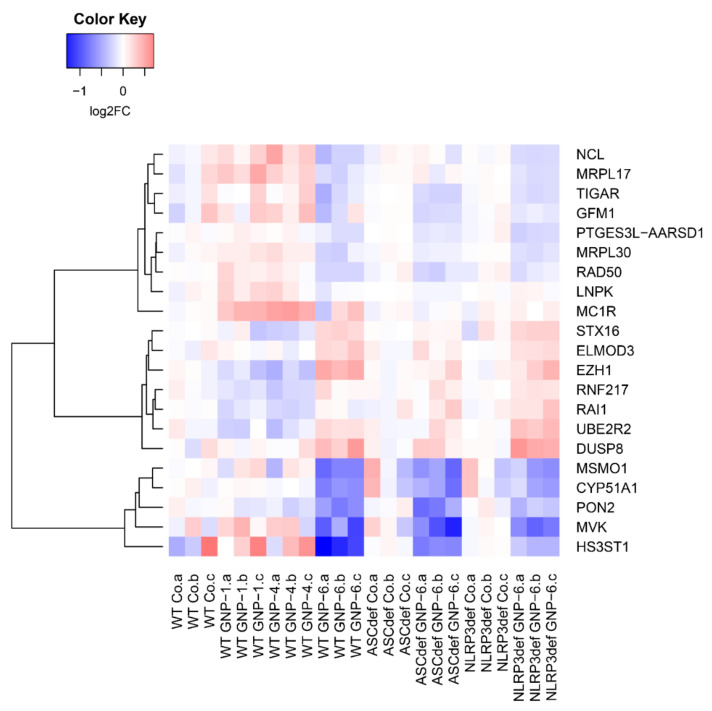
Heatmap of a selection of the 103 DEGs showing each individual response compared to its own respective control. Individual samples are shown. The selection was made to contrast effects on GNP-1 and GNP-4 vs. GNP-6. Co: unexposed controls; log2FC: ^2^log of the fold change.

**Table 1 ijms-23-05763-t001:** Characteristics of the gold nanoparticles used. COOH-PEG(3kDa)-SH, alpha-Thio-omega-carboxy poly(ethylene glycol); MeO-PEG(5kDa)-SH, alpha-Methoxy-omega-mercapto poly(ethylene glycol).

GNPs	Shape	Size	Surface Modification
1	star	60 nm	COOH-PEG(3kDa)-SH
2	star	44 nm	COOH-PEG(3kDa)-SH
3	sphere	30 nm	MeO-PEG(5kDa)-SH
4	sphere	50 nm	MeO-PEG(5kDa)-SH
5	rod	40 nm × 16 nm	MeO-PEG(5kDa)-SH
6	rod	60 nm × 14 nm	MeO-PEG(5kDa)-SH

**Table 2 ijms-23-05763-t002:** Endotoxin content of 100 µg/mL GNP dispersions. A, below 0.05 EU/mL threshold; B, above 0.05 EU/mL threshold; C, accurate measurement is precluded by GNP interference.

GNPs	Endotoxin (EU/mL)	Outcome
1	0.030	A
2	0.070	B
3	1.667	C
4	1.546	C
5	0.436	C
6	0.041	A

**Table 3 ijms-23-05763-t003:** Concentrations inducing a 10% loss of viability (10% Effective Dose, or ED_10_, including a 90% Confidence Interval (CI)), calculated by dose–response modelling. For nanostars, an ED_10_ could not be calculated. The data of the three replicate experiments were analysed together.

	Nanospheres		Nanorods	
	ED_10_ (µg/mL)	90% CI	ED_10_ (µg/mL)	90% CI
Wildtype	9.5	2.4–18.0	34.1	20.4–51.8
ASC-def	12	6.8–17.1	169	97.2–299
NLRP3-def	17.7	10.7–18.5	166	47.7–242

**Table 4 ijms-23-05763-t004:** Number of differentially expressed genes (DEGs) for (A) GNP treatments vs. unexposed controls (C) in wildtype, ASC-deficient, and NLRP3-deficient THP-1-derived cells, and (B) unexposed wildtype vs. ASC-deficient and NLRP3-deficient THP-1-derived cells. *p* < 0.001. False discovery rate (FDR) = 11%. The total number of DEGs in panel A is 103 since, from the 40 DEGs for NLRP3-def, one overlaps with GNP-4 nanospheres vs. C (wildtype) and one with GNP-6 nanorods vs. C (ASC-def).

Comparison	Cells	# of DEGs
**A**		
GNP-1 nanostars vs. C	Wild-type	6
GNP-4 nanospheres vs. C	Wild-type	9
GNP-6 nanorods vs. C	Wild-type	15
GNP-6 nanorods vs. C	ASC-def	35
GNP-6 nanorods vs. C	NLRP3-def	40
**B**		
Wildtype C vs. ASC-def C		332
Wildtype C vs. NLRP3-def C		219

## Data Availability

Upon manuscript acceptance, the data will be deposited in a publicly available database.

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
