# Peer review of "Pathways Related to NLRP3 Inflammasome Activation Induced by Gold Nanorods"

_ijms, 2022, doi:10.3390/ijms23105763_

Round 1

Reviewer 1 Report

In this manuscript, the authors report that gold nanorods, but not nanostars or nanospheres, trigger NLRP3 inflammasome activation in differentiated THP-1 cells. Indeed, increased concentrations of nanorods promote cell death associated with a release of the pro-inflammatory cytokines IL-1b and IL-18. Using ASC- or NLRP3-deficient THP1 cells, IL-1b production was completed abrogated following nanorods treatment confirming the involvement of the NLRP3 inflammasome for the production of this cytokine (Fig 3). Surprisingly, the loss of ASC or NLRP3 had only a marginal effect on the cell death suggesting that another form of cell death, in parallel to pyroptosis, is likely induced by the nanorods. Have the authors assessed whether nanorods promote apoptosis? Nanorods may affect the mitochondrial integrity leading to cytochrome c release. It is a hypothesis that should be explored.

Apoptotic effector caspases as caspase-3 or -7 have been reported to activate ribonucleases targeting some groups of RNAs. Here, as the authors observed that gold nanorods induce a downregulation of sterol/cholesterol biosynthesis as well as in oxidative phosphorylation, this possibility should be kept in mind. As mentioned by the authors, those downregulations are NLRP3 inflammasome independent since they are also observed in ASC or NLRP3 deficient cells. That’s a bit puzzling.

Other comments

In Fig 2, in the presence of nanorods, the production of IL-1b is about 200% of the control which is not that strong while in Fig 3, about 5 times more Il-1b are produced in nanorods treated cells. How do the authors explain the differences between these similar experiments?

I’m sorry but I could not find in which figure is shown the upregulation of the purinergic receptor upon nanorods treatment.

Author Response

dear reviewer,

kind regards

Rob Vandebriel, PhD

Reviewer 2 Report

The authors investigated about the effects of gold ENM of different shapes on NLRP3 inflammasome activation and related signalling pathways.

The rational behind the study was clear and straight forward. The manuscript is almost well written.Overall the topic could be interesting but many details are not clear.

 I recommend that the paper be accepted with minor revision:

a) Ii the abstract section, the authors should emphasize  the conclusions of their study.

b) In the introduction section, little previous evidence is provided about the importance of NLRP3 activation in inflammatory diseases. Incorporating comparisons with other studies would increase the strength of the paper. Please refer to doi: 10.3390/ijms21062144; 10.1083/jcb.202006194; 10.3390/antiox9100992.

c) The authors should better describe the conclusions.

d) There are some minor grammar issues that should be fixed in order to aid the accessibility of the results to the reader.

Author Response

(The authors gave the same response as above.)

Round 2

Reviewer 1 Report

To assess whether apoptosis may be triggered by Nanorods, authors performed qPCR of caspase-1, -10 or cytochrome c. First, those caspases are poorly involved in apoptosis unlike caspase-3. Moreover, caspase up-regulation is not required "per se" for their activation. Likewise, cytochrome c promotes caspase activation after its release into the cytosol, not because it is up-regulated. Why don't the authors assess a lack of caspase-3 activity (or the absence of cytochrome c release) in their model to really rule out the involvement of apoptosis? 

Author Response

We agree with the reviewer that measuring caspase-3 activity is an important control. However, upon exposure the cells were directly frozen for RNA isolation, and not previously lysed to enable measurement of caspase-3. Since the project is finished we do not have funds to set up a new triplicate experiment to perform this control. In the revised manuscript we added: “The study did not investigate the mechanisms underlying the cell death observed. Most importantly, a possible role of apoptosis could be evaluated by measuring caspase-3 activity.”

Round 3

Reviewer 1 Report

I recommend acceptance for publication

Author Response

thanks.